Campsites, forest fires, and entry point distance affect earthworm abundance in the Boundary Waters Canoe Area Wilderness

Wellnitz Todd wellnita@uwec.edu
Barlow Jenna L.
Dick Cory M.
Shaurette Terrance R.
Johnson Brian M.
Wesley Troy
Weiher Evan
Biology Department, University of Wisconsin-Eau Claire , Eau Claire, WI , USA
Robinson Andrew
Electronic publication date: 2020 Feb 25
Publication date: 2020
Volume: 8
Electronic Location ID: e8656
Received 2019 Mar 19; Accepted 2020 Jan 28
Copyright: © 2020 Wellnitz et al.
Copyright year: 2020
Copyright holder: Wellnitz et al.
License: This is an open access article distributed under the terms of the Creative Commons Attribution License, which permits unrestricted use, distribution, reproduction and adaptation in any medium and for any purpose provided that it is properly attributed. For attribution, the original author(s), title, publication source (PeerJ) and either DOI or URL of the article must be cited.
License URL: https://creativecommons.org/licenses/by/4.0/

Keywords: Fire, Campsites, BWCAW, Entry point, Lake islands, Human impacts, Invasive species, Earthworms

Funding: University of Wisconsin—Eau Claire’s Office of Research and Sponsored Programs This project was supported by BluGold Commitment funding provided by the University of Wisconsin—Eau Claire’s Office of Research and Sponsored Programs. The funders had no role in study design, data collection and analysis, decision to publish, or preparation of the manuscript.

==============================
Factors controlling the spread of invasive earthworms in Minnesota’s Boundary Waters Canoe Area Wilderness are poorly known. Believed to have been introduced by anglers who use them as bait, invasive earthworms can alter the physical and chemical properties of soil and modify forest plant communities. To examine factors influencing earthworm distribution and abundance, we sampled 38 islands across five lakes to assess the effects of campsites, fire and entry point distance on earthworm density, biomass and species richness. We hypothesized that all three parameters would be greater on islands with campsites, lower on burned islands and would decrease with distance from the wilderness entry point. In addition to sampling earthworms, we collected soil cores to examine soil organic matter and recorded ground and vegetation cover. Campsite presence was the single most important factor affecting sampled earthworm communities; density, biomass and species richness were all higher on islands having campsites. Fire was associated with reduced earthworm density, but had no direct effects on earthworm biomass or species richness. Fire influenced earthworm biomass primarily through its negative relationship to groundcover and through an interaction with entry point distance. Entry point distance itself affected earthworm density and biomass. For islands with campsites, earthworm biomass increased with distance from the entry point.

Introduction

Following the end of the last glaciation 12,000 years ago, northern forests of the present-day USA were believed to have been entirely free of earthworms (Tiunov et al., 2006). Since the glacial retreat, native earthworm populations have largely remained below the glaciers’ southern terminus, but in recent times non-native earthworms from Europe have proliferated and become increasingly common on old glaciated landscapes (Callaham et al., 2006; Hopfensperger & Hamilton, 2015). As non-native earthworms expanded their range north, they have profoundly impacted the northern forest ecosystems that evolved in their absence (Hale, Frelich & Reich, 2005; Frelich et al., 2006). Many earthworms feed on leaf litter and decomposing plant material that accumulates as a spongy layer of duff on the forest floor. The duff layer provides habitat for ground-dwelling animals and understory vegetation and helps prevent soil erosion. Earthworms consume the duff and excrete nutrient-rich castings, which in combination with their bioturbation and mixing of soil layers, can stimulate microbial activity and greatly accelerate soil nutrient cycling (Bohlen et al., 2004a). Although these processes benefit horticultural and agricultural systems (Bertrand et al., 2015; Sharma, Tomar & Chakraborty, 2017), they can fundamentally alter the functioning of northern forest ecosystems (Bohlen et al., 2004b). Among other detrimental effects, earthworms can alter forest seed banks, kill plant roots and increase soil compaction, which can lead to increased erosion and nutrient leaching (Costello & Lamberti, 2009; Hopfensperger, Leighton & Fahey, 2011; Drouin et al., 2014; Nuzzo, Davalos & Blossey, 2015) and negatively affect native flora and fauna (Hale et al., 2008; Loss & Blair, 2011; Fisichelli et al., 2013; Dobson & Blossey, 2015).

Non-native earthworms are common in the Boundary Waters, a vast wilderness region of interconnected lakes straddling the Canada–United States border between Ontario and Minnesota (Fig. 1). In northern Minnesota this region has been designated the Boundary Water Canoe Area Wilderness (BWCAW). Known for its pristine lakes and excellent sports fishing, the BWCAW has long been a destination for anglers (Heinselman, 1999), and it is probable that anglers played an important role in spreading non-native earthworms into the area by using them as fish bait (Cameron, Bayne & Clapperton, 2007; Hale, 2008; Kilian et al., 2012). Non-native earthworms such as “night crawlers” (Lumbricus terrestris) and “red wigglers” (Lumbricus rubellus or Dendrodrilus rubidus) have been sold as fishing bait for decades and are now abundant in BWCAW forests. It is not unusual for anglers to dump unused bait (Keller et al., 2007; Frelich & Reich, 2009), and even if the discarded earthworm adults are dead, the soil in which they are packed may contain viable earthworm cocoons that could invade a site. Campsites are likely dumping grounds for unused bait and probable points of earthworm invasion because visitors to the BWCAW are required to stay at designated campsites.

Figure 1 Study location.

Map shows the location of the Boundary Waters Canoe Area Wilderness in northern Minnesota (inset) and the study area (small white rectangle), which has been enlarged to show the six connecting lakes (One, Two, Three, Four, Hudson and Insula) and the 38 islands that were sampled (shown in black). The dotted line represents the northern extent of the 2011 Pagami Creek Fire; the area south of the line was burned by the fire. Image credit: Map base courtesy of Bruce Jones Design and FreeUSandWorldMaps.com.

While earthworms are novel to the BWCAW, fire has shaped the region for millennia (Heinselman, 1999). Historically, fire disturbance occurred at 50–100 year intervals, but fire suppression practices over the past century have increased fire intervals to 700 years or more (Heinselman, 1999; Frelich & Reich, 2009). Forest fire frequency could matter to earthworms because burning away duff and litter layers has the potential for controlling their numbers (Callaham et al., 2003). Fire kills adults that inhabit the leaf litter (i.e., epigeic species) and may reduce the viability of earthworm cocoons in the soil (Ikeda et al., 2015). For soil-dwelling and deep-burrowing earthworms (i.e., endogenic and anecic species), burning away the duff and litter layers would limit food resources, likely starve earthworms and reduce their fitness (Callaham & Blair, 1999; Coyle et al., 2017). Despite its potential for earthworm control, only a handful of studies have examined how fire affects earthworm dynamics in North American ecosystems (James, 1982, 1988; Callaham & Blair, 1999; Callaham et al., 2003; Ikeda et al., 2015) and fewer still have examined earthworm responses to fire in northern forests (Frelich et al., 2006).

To assess the role of campsites and forest fires in influencing the distribution and abundance of non-native earthworms in the BWCAW, we examined lake islands that either had or did not have campsites and were either burned or not burned by the Pagami Creek Fire of 2011. Earthworm abundance was measured in terms of earthworm density, biomass and species richness. We studied islands because they constitute discrete forest patches separated by water that could be characterized as having one condition or the other for each variable (i.e., campsites present/absent, burned/not burned). The islands sampled were located along a chain of interconnected lakes that bordered the northern edge of the 2011 Pagami Creek Fire (Fig. 1). Canoe access for this lake chain is largely limited to a single entry point on Lake One at the west end of the chain. This isolated entry allowed us to examine how distance from the source pool—that is, arriving anglers carrying earthworm bait—affected earthworm abundance across the islands examined.

We hypothesized that islands with campsites would have more earthworms than those without because campsites would receive multiple earthworm introductions from anglers dumping unused bait in or near the campsite (Novo et al., 2015). By contrast, islands that were burned would have fewer earthworms than those not burned because decreased duff and litter on the forest floor, reduced soil organic matter and decreased leaf litter inputs from trees and vegetation would decrease food availability, thereby limiting earthworm growth and fecundity. Islands having the greatest number of earthworms, therefore, would be those that had campsites and were not burned, whereas those having the least would be burned and without campsites. We also hypothesized that earthworm density and biomass would decrease as distance from the entry point increased. We assumed campsites closest to the entry point would have higher visitation rates because day and weekend anglers would be less likely to travel far into the wilderness (Lime, 1971). Hypothesized relationships between variables and earthworm biomass are summarized in Fig. 2.

Figure 2 Metamodel depicting hypothesized relationships.

Solid lines show positive relationships; dashed lines show negative relationships.

Field-site description

The 4,410 km2 BWCA Wilderness is located in Minnesota’s Superior National Forest (47°57′ N, 91°48′ W) and extends 185 km along the Minnesota/Ontario border (Fig. 1, inset). Designated as a wilderness in 1978, the area contains approximately 1,175 glacial lakes and hundreds of kilometers of rivers and streams (Heinselman, 1999). It is the largest Forest Service Wilderness area east of the Rocky Mountains and the most heavily used wilderness in the United States, with over 150,000 overnight and day-use visitors annually (Eagleston & Marion, 2017a). Lakes and waterways comprise approximately 20% of the BWCAW and cross-country canoe travel is made possible by a network of portage trails that connect lakes. Forests in the BWCAW are a mixture of conifers and hardwoods that include various species of pine (Pinus banksiana, P. resinosa and P. strobus), white cedar (Thuja occidentalis), balsam fir (Abies basamea), white spruce (Picea glauca), and hardwoods such as paper birch (Betula papyrifera), big-toothed aspen (Populus grandidentata) and white oak (Quercus alba) (Dickens, Gerhardt & Collinge, 2005).

The climate of the BWCAW region is hemiboreal with mild/cool summers and long winters (Köppen classification Dfb). Average July and January temperatures are 20 °C and −11 °C, respectively, with temperature extremes of 32 °C and −40 °C occurring (Dickens, Gerhardt & Collinge, 2005). Annual precipitation ranges 66–78 cm and approximately 40% of this occurs as winter snow. Soils are thin and acidic and overlay Canadian Shield bedrock. Charcoal is almost universal in upper soil layers, indicative of the key role fire has played in shaping this ecosystem (Heinselman, 1999).

Recreational access to the BWCAW is controlled through a permit-based quota system that limits the number of people entering through each wilderness entry point. Visitors can obtain permits from May to September and group sizes are limited to nine. Camping is restricted to approximately 2,000 designated campsites established by the by U.S. Forest Service, that are unambiguously indicated by cast iron fire grates located at each site (Eagleston & Marion, 2017b).

Materials and Methods

On 9 July 2016 we put in canoes at BWCAW entry point 30 on Lake One and headed east along a route that skirted the northern edge of the 2011 Pagami Creek Fire (Fig. 1). Over 7 days we sampled 38 islands (Boundary Waters Canoe Area Wilderness permit #6-2861367) on lakes One, Two, Three, Four, Hudson and Insula. Islands were selected on the basis of their fire history (burned/not burned) and the presence/absence of campsites (Table 1). Island selection was made using maps showing campsite locations (Fisher map F4) and the area covered by the Pagami Creek Fire (USDA Pagami Creek Fire Map). Initial selection followed a balanced design for each combination of characteristics (i.e., burned/campsite, burned/no campsite, not burned/campsite, not burned/no campsite). However, field sampling revealed islands marked as burned were not always so, and some campsites on burned islands could not be found. Consequently, these islands were discarded from our sample set and this resulted in an unbalanced design.

Table 1 Physical characteristics of the 38 sampled islands.

“X”s indicate that islands were burned by the Pagami Creek Fire or had a campsite. Entry distance is the linear distance from BWCAW entry point 30 on Lake One (see Fig. 1).

Island #	Latitude	Longitude	Lake	Burned	Campsite	Entry distance (km)	Island area (ha)	
1	N 47 55.45	W 91 29.10	One	–	X	3.4	3.85	
2	N 47 55.31	W 91 29.19	One	–	–	3.7	4.72	
3	N 47 55.17	W 91 29.68	One	–	–	4.1	2.89	
4	N 47 55.27	W 91 29.56	One	–	X	4.2	0.94	
5	N 47 55.03	W 91 29.21	One	–	–	4.3	0.52	
6	N 47 54.90	W 91 29.62	One	–	–	4.4	2.12	
7	N 47 55.20	W 91 29.71	One	–	–	4.5	2.16	
8	N 47 55.03	W 91 28.99	One	–	–	4.5	0.72	
9	N 47 55.03	W 91 29.21	One	–	–	4.8	0.83	
10	N 47 54.33	W 91 27.79	Two	X	X	6.7	1.40	
11	N 47 54.43	W 91 27.33	Two	–	–	7.0	0.60	
12	N 47 53.04	W 91 26.69	Three	–	–	7.2	0.40	
13	N 47 53.81	W 91 26.60	Three	–	X	8.8	2.63	
14	N 47 53.58	W 91 26.73	Three	–	–	9.0	0.53	
15	N 47 53.70	W 91 26.56	Three	X	–	9.0	0.75	
16	N 47 53.42	W 91 27.09	Three	–	–	9.1	0.65	
17	N 47 53.02	W 91 26.50	Three	X	X	9.4	6.58	
18	N 47 53.04	W 91 26.94	Three	X	–	9.5	11.81	
19	N 47 52.98	W 91 27.11	Three	X	X	10.0	9.25	
20	N 47 53.75	W 91 25.85	Three	–	–	10.7	5.96	
21	N 47 53.01	W 91 26.52	Three	X	–	10.8	1.24	
22	N 47 53.12	W 91 26.53	Three	–	X	11.0	0.73	
23	N 47 54.00	W 91 23.11	Four	–	–	14.0	1.74	
24	N 47 53.73	W 91 20.85	Hudson	X	X	17.4	1.31	
25	N 47 54.24	W 91 17.02	Insula	X	–	23.0	5.02	
26	N 47 54.22	W 91 16.38	Insula	X	X	23.6	1.21	
27	N 47 54.56	W 91 16.82	Insula	–	X	24.0	2.28	
28	N 47 54.55	W 91 17.06	Insula	X	–	24.0	0.64	
29	N 47 54.62	W 91 17.41	Insula	X	–	24.0	0.52	
30	N 47 54.70	W 91 17.23	Insula	–	X	24.1	4.17	
31	Missing	Missing	Insula	–	–	24.1	7.31	
32	N 47 54.71	W 91 17.64	Insula	X	–	24.3	1.18	
33	N 47 54.79	W 91 17.77	Insula	–	–	24.6	0.45	
34	N 47 55.16	W 91 18.05	Insula	–	–	25.4	0.68	
35	N 47 55.65	W 91 16.80	Insula	X	X	25.7	0.90	
36	N 47 55.65	W 91 17.02	Insula	–	–	25.9	1.27	
37	N 47 55.79	W 91 17.26	Insula	–	–	26.1	0.71	
38	N 47 55.93	W 91 17.11	Insula	–	X	26.3	1.28	

Distances between islands and the entry point were examined using straight-line distances instead of hypothetical canoe travel distances because the latter can be highly variable. Canoeists and anglers need not follow direct pathways between lakes and islands and side trips to explore fishing spots and campsite locations are common. Also, multiple pathways may exist for reaching islands, especially in geographically complex lakes such as Insula. Thus, determining actual travel routes becomes problematic whereas straight-line distances are easily made and independent of human choice.

Sampling islands followed a standard procedure: Upon reaching an island, its burn condition, campsite status and GPS coordinates were recorded. Following this, three sampling sites spaced equidistantly around the island’s perimeter and approximately 10 m inland were chosen using three criteria: (1) site accessibility from shore (2) availability of flat and open ground and (3) lack of dense underbrush allowing easy access to the ground for sampling. After reaching a site, a center point was established and the percent of understory vegetation cover and percent of exposed soil was estimated within a 1 m radius of the site center. Visual estimates of cover and exposure were made with the aid of a chart depicting the Braun–Blanquet percent cover classes. Percent exposed soil was converted into percent groundcover by subtracting values from 100. To assess soil organic matter, three soil cores were collected within the established 1 m radius. The upper 10 cm of each 2.5 cm diameter core was collected and combined in a plastic bag for later organic matter analysis in the lab.

Earthworms were sampled following the procedures outlined in Hale (2013). A mustard water solution was prepared by combining 240 ml of powdered mustard with 4 L of water. This solution acts as an irritant that when poured on the ground and allowed to soak into the soil causes earthworms to emerge as they attempt to avoid the mustard. Half of this mustard solution was poured inside a 0.2 m2 quadrat placed at the site’s center and emerging earthworms were collected with forceps and placed in 50 ml plastic centrifuge tubes. After 2 min, the remaining solution was poured into the quadrat and earthworms were collected until they stopped emerging. This procedure was repeated at each site and earthworms from all three sites were combined for each island and preserved in a solution of 70% isopropyl alcohol and 2% formalin. Field conditions made it impractical to clear earthworm guts prior to preservation. Back in the laboratory, earthworms were identified using Hale (2013) and counted. Worms from each site were then combined, dried and weighed to determine biomass.

We used the loss-on-ignition method to determined soil organic matter. Soil was dried in an oven at 40° C for 7 days, then homogenized with mortar and pestle. Homogenized samples were put into pre-weighed porcelain crucibles and weighed, then ignited in a muffle-furnace (Isotemp Muffle Furnace Model 550-126; Fisher Scientific, Hampton, NH, USA) at 550 °C for 3 h. The samples were subsequently cooled, placed in a desiccator overnight and then weighed to the nearest 0.001 g so biomass could be calculated.

Statistical analyses

Structural Equation Modelling (SEM) was used to examine relationships between earthworm biomass and the variables included in the metamodel (Fig. 2). To examine the effect of campsite presence, fire history and entry point distance on earthworm richness, density and biomass, we used General Linear Models (GLM).

The SEM was created in AMOS 16 (2007; Amos Development Corporation, Meadville, PA, USA) and used to examine direct and indirect relationships between earthworm biomass and fire history, entry point distance, understory plant cover, groundcover and soil organic matter. We used four indices to assess the fit of the SEM to avoid bias from any one index (Hintz & Lonzarich, 2018). A chi-square test evaluated the consistency of the data, with p-values greater than 0.05 specifying good fit (Grace et al., 2010). To evaluate model fit we looked at the goodness-of-fit index (GFI), standardized root mean square residual (SRMR) and root mean square error of approximation (RMSEA). GFI values greater than 0.90 and SRMR and RMSEA values below 0.08 indicate good fit (Hooper, Couglan & Mullen, 2008; Hu & Bentler, 1999).

In order to investigate a possible lack of independence among samples collected from the same lake, we calculated the residuals from the SEM and compared them across lakes. The resulting boxplot showed no evidence of patterns across the lakes. We also plotted a semivariogram of the residuals to assess possible spatial autocorrelation and no evidence for this was found. Thus, samples from the same lake were treated as independent of one another.

GLMs were performed using JMP 8.0.1 (2009; SAS Institute Inc., Cary, NC, USA). Models used a Poisson distribution to fit variables and a log link function to create linear models. Chi-squared (X2) probabilities ascertained significance levels of individual factors and interactions. We started with a full model that included main factor effects and all interactions, but found that three-way interactions were not significant so discarded them from the final models. We did not use a nested design (i.e., islands nested within lakes) because the SEM residual analysis showed no lake effect.

Results

We sampled 38 islands across the six connecting lakes, of these, 12 were burned and 13 had campsites (Fig. 1; Table 1). Island size ranged from 0.4 to 11.8 ha and straight-line distances from the entry point ranged between 3.4 and 26.3 km. Earthworms occurred on all but two of the sampled islands. Mean (±1 SE) earthworm biomass and density across islands was 12.2 ± 2.5 g m−2 and 27.2 ± 3.5 individuals m−2, respectively.

Eight earthworm species were identified (Table 2), of which Dendrobaena octaedra (Savigny, 1826) and L. rubellus (Hoffmeister, 1843) were most common. Species richness on islands averaged 1.95 species. Juvenile Lumbricus spp. were the most frequently encountered taxonomic group and many of these juveniles were likely L. terrestris (Linnaeus, 1758); however, their developmental stage precluded positive identification. Campsite presence was the sole factor affecting earthworm species richness (X2 = 4.13, df = 1, 37; p = 0.04), with campsite-bearing islands averaging 1.13 more species than those without.

Table 2 Earthworm species or juvenile genera encountered on sampled islands.

“X”s indicate that islands were burned by the Pagami Creek Fire or had a campsite.

Island #	Burned	Campsite	Eisenia fetida	Dendrobaena octaedra	Dendrodrilus terrestris	Lumbricus rubellus	Allolobophora chlorotice	Aporrectodea rosea	Juvenile Aporrectodea	Octolasion tyrteum	Lumbricus terrestris	Juvenile Lumbricus	
1	–	X	5	–	–	–	–	–	–	–	–	–	
2	–	–	–	45	–	–	–	–	–	–	–	–	
3	–	–	–	–	–	–	–	5	–	–	–	20	
4	–	X	–	–	–	–	–	–	–	–	–	5	
5	–	–	–	30	–	–	–	–	–	–	–	–	
6	–	–	–	10	–	–	–	–	–	–	–	–	
7	–	–	–	30	–	–	–	–	–	–	–	–	
8	–	–	–	5	–	–	–	5	–	–	–	35	
9	–	–	–	5	–	–	–	–	–	–	–	–	
10	X	X	–	–	–	–	–	–	5	–	–	20	
11	–	–	–	30	–	–	–	–	–	–	–	–	
12	–	–	5	5	–	–	–	–	–	–	–	–	
13	–	X	–	–	–	30	–	–	10	–	–	50	
14	–	–	–	–	–	15	–	–	5	–	–	35	
15	X	–	–	–	–	–	–	–	–	–	5	20	
16	–	–	–	10	–	5	–	–	–	–	–	–	
17	X	X	10	5	–	–	–	–	–	5	5	–	
18	X	–	–	–	–	10	–	–	–	–	–	10	
19	X	X	–	–	–	5	–	–	–	–	5	10	
20	–	–	–	25	–	–	–	–	–	–	–	–	
21	X	–	–	–	–	10	–	–	–	–	–	35	
22	–	X	40	5	–	–	5	–	5	10	–	10	
23	–	–	–	5	–	–	–	–	–	–	–	5	
24	X	X	–	–	–	–	–	–	–	–	5	10	
25	X	–	–	–	–	–	–	–	–	–	–	–	
26	X	X	–	–	–	–	–	–	–	–	–	10	
27	–	X	–	5	–	–	–	–	–	–	5	20	
28	X	–	–	10	–	–	–	–	–	–	–	–	
29	X	–	–	5	–	–	–	–	–	–	–	5	
30	–	X	–	–	10	–	–	5	15	–	–	30	
31	–	–	–	15	–	–	–	–	–	–	–	10	
32	X	–	–	–	–	–	–	–	–	–	–	15	
33	–	–	–	–	–	–	–	–	–	–	–	–	
34	–	–	–	20	–	–	–	–	–	–	–	10	
35	X	X	–	–	–	–	–	–	5	–	5	30	
36	–	–	–	–	–	–	–	10	–	–	–	–	
37	–	–	–	–	–	–	–	–	–	–	–	15	
38	–	X	–	–	–	20	–	–	–	–	5	–	
Totals			60	265	10	95	5	25	45	15	35	410	

Campsite presence and burn history had important effects on earthworm abundance. Campsite presence was the only factor that affected both earthworm biomass and density (Table 3). The effect on biomass was greater, with campsite-bearing islands having 81% more earthworm biomass as compared to islands lacking campsites. By comparison, earthworm density on campsite-bearing islands increased by only 37%. Fire was associated with reductions in earthworm density, but not biomass (Table 3; Fig. 3). Earthworm density also responded to a Burned × Campsite interaction such that burned, campsite-bearing islands had approximately half the density of non-burned, campsite-bearing islands (Fig. 3A). Islands without campsites had comparable earthworm densities regardless of burn history.

Figure 3 Interactive effects influencing earthworm biomass.

The effect of campsite presence and fire history on earthworm biomass and density. Values are mean (±1 SEM). GLM analyses indicated a significant campsite × burn interaction for earthworm density (A), but not earthworm biomass (B).

Table 3 Effects of campsites, fire and entry point distance on earthworm biomass and density.

Results of the two General Linear Models used to examine earthworm biomass (g m−2) and density (individuals m−2) in relation to burn history (Burned), campsite presence (Campsite) and linear distance from the BWCAW entry point (Distance), as well as interactions among these factors (df = 1, 36). Bold font indicates significant p-values. Whole model statistics: X2 density = 45.30, df = 6, 31, p < 0.0001; X2 biomass = 84.23, df = 6, 31, p < 0.0001.

Factor	Biomass	Density	
X2	p-Value	X2	p-Value	
Burned	0.67	0.413	8.51	0.004	
Campsite	27.82	<0.001	4.92	0.027	
Distance	0.45	0.500	17.01	<0.001	
Burned × Campsite	0.19	0.665	13.74	<0.001	
Campsite × Distance	5.32	0.021	1.25	0.26	
Distance × Burned	6.41	0.011	2.19	0.14	

Distance from the entry point directly influenced earthworm density and had interactive effects with campsite presence and burn history on earthworm biomass (Table 3; Fig. 4). Across islands, earthworm density showed a significant decrease as entry point distance increased. By contrast, earthworm biomass increased with entry point distance for campsite-bearing islands while remaining unchanged for islands without campsites (i.e., the Campsite × Distance interaction shown in Fig. 4A). There was also a Distance × Burned interaction such that earthworm biomass decreased as distance from the entry point increased on burned islands, whereas non-burned islands showed the opposite trend (Fig. 4B).

Figure 4 Interactive effects influencing earthworm biomass.

The interaction between entry point distance and burn history (A), and entry point distance and campsite presence (B) on earthworm biomass.

Our SEM (Fig. 5) showed a good fit to the data with a high GFI (0.96), low SRMR (0.08) and RMSEA (0.00) and a non-significant chi-square (X2 = 6.32, df = 9, p = 0.71). Overall, the model explained 47% of the variation in earthworm biomass found on BWCAW islands.

Figure 5 Structural equation model representing the relationships between study variables and earthworm biomass.

Latent variables associated with endogenous factors are not shown. Solid arrows indicate positive effects; dashed arrows indicate negative effects. Dotted gray arrows labeled “ns” indicated non-significant relationships (p > 0.10). Model parameters are as follows: chi-square = 6.32, df = 9, p = 0.71; GFI = 0.96; RMSEA = 0.00; SRMR = 0.081.

SEM analysis broadly corroborated the results of the GLM in that both models showed that campsite presence was the single most important factor affecting earthworm biomass. The SEM showed that entry point distance had a direct and positive relationship with earthworm biomass whereas fire exerted its influence through other factors. Fire was associated with less groundcover and soil organic matter, but more understory plant cover. Of these three factors, only groundcover affected earthworm biomass. Groundcover had a positive relationship to earthworm biomass whereas soil organic matter and understory plant cover had none. Understory plant cover was also positively correlated to earthworm biomass.

Discussion

We hypothesized that earthworms would be more abundant on islands with campsites, less abundant on those that had burned and would decrease as distance from the entry point increased. Our data provide strong support for the first hypothesis (campsites), partial support for the second (burn history) and little support for the third (entry point distance).

Whether measured in terms of biomass or density, earthworms were consistently more abundant on islands with campsites as compared to those without. This result corroborates the findings of previous studies showing that human activity influences the distribution and abundance of non-native earthworms (Hendrix et al., 2008; Shartell et al., 2015). These include correlations with road proximity (Cameron & Bayne, 2009) and boat launches (Cameron, Bayne & Clapperton, 2007) and their spread through use as fish bait (Keller et al., 2007). That wilderness campsites may be important points of invasion for non-native species is often overlooked in studies of campsite impacts (e.g., Monz, Pickering & Hadwen, 2013; Eagleston & Marion, 2017b; Marion et al., 2016), despite the fact that campsites are where people leave behind food wastes (a potential source of seeds) and may inadvertently transport seeds and other propagules on clothing, footwear or camping equipment (Ansong & Pickering, 2013; Eagleston & Marion, 2017a). Campsites might be construed, therefore, as nodes in a distribution network facilitating the spread of exotic species across wilderness areas. This may be especially true in the BWCAW because nearly all travel is done by canoe, which makes established campsites and canoe portages (Dickens, Gerhardt & Collinge, 2005) the main points of contact between visitors and the wilderness landscape.

Fire effects were not as prominent as campsite effects. Although burned islands did have lower densities of earthworms as predicted, earthworm biomass and richness were unaffected. Fires can kill epigeic species that live in the litter (Ikeda et al., 2015), but are less likely to affect endogenic and anecic species that burrow into the soil. Fire increases soil temperatures just a few degrees 5 cm below the surface (Ikeda et al., 2015), and some earthworms could burrow deeper to avoid excessive heat. Rather than killing worms directly, we expected fire to exert influence through its effects on ground cover, soil organic matter and understory vegetation, decreasing the former two and increasing the latter. Our SEM showed that fire had the predicted effects on these three variables, but only groundcover had a significant effect on earthworm biomass such that, where litter was abundant, so was earthworm biomass.

Less clear is why understory vegetation responded positively to earthworms. Numerous studies have shown that earthworms invading northern forests cause understory vegetation to decline (see Bohlen et al., 2004b for review). Earthworms can consume newly germinated seedlings (Drouin et al., 2014), the fine roots of plants (Gilbert et al., 2014), and alter the soil microflora to favor only certain species (Drouin, Bradley & Lapointe, 2016). Of course, earthworms can have beneficial effects, as is seen in horticultural and agricultural systems (Bertrand et al., 2015; Sharma, Tomar & Chakraborty, 2017), but why these benefits would occur on islands and not in other areas of the BWCAW is outside the scope of our study.

The most perplexing result of our study was the positive relationship between entry point distance and earthworm biomass. Our prediction that earthworm biomass would decrease on islands further away from the entry point was based on the assumption that discarding unused earthworm bait would be more frequent in areas of high human traffic closer to the entry point. We reasoned that weekend anglers do not travel far from entry points and bait dumping would be less frequent as angler traffic diminished further away. Of course, our assumptions about human behavior may be wrong. We speculate that anglers may actually be less likely to dispose of worms if they are fishing for only short time. In a survey of Wisconsin and Michigan boat owners, Keller et al. (2007) reported that 65% of anglers saved leftover earthworm bait for future fishing trips as opposed to 41% who disposed of them on land or in trash. Similar results were found by Kilian et al. (2012) in their survey of Maryland anglers. Earthworms are sold in soil-packed containers and can be kept alive for weeks under cool and moist conditions (Sherman, 2003). If bait containers are exposed to long days in a sunlit canoe, however, their viability may be compromised. Desiccating soil and temperatures exceeding 30 °C can kill earthworms (Berry & Jordan, 2001), and we speculate that multi-day canoe trips are likely to have higher bait mortality than day or weekend trips. Rather than pack out dead and dying worms, long-distance anglers may choose to dump bait containers enroute as they pass through the wilderness. Even if adult earthworms in containers are dead, the soil in which they were kept could contain earthworm cocoons. Whether or not this practice actually occurs would require further study, yet it highlights the need for understanding human behavior for managing the spread of earthworms in the BWCAW (Eagleston & Marion, 2017a, 2017b; Marion et al., 2016).

Conclusion

Earthworms are often called ecosystem engineers because of the multitudinous effects they have on soils, nutrient cycling and plants and animals (Holdsworth, Frelich & Reich, 2007; Eisenhauer, 2010). Their presence can profoundly alter ecosystems and their invasion into wilderness areas presents a challenge for resource managers charged with maintaining these natural systems in a state similar to what they were pre-European settlement (Blouin et al., 2013). Our investigation of how campsites, fire and entry point proximity influence earthworm abundance in the BWCA has management implications. Our data support the hypothesis that campsites are key points of invasion and suggest that fire may lower earthworm densities via its influence on worm food resources (e.g., litter and duff). Our data also indicate that the distance from a wilderness entry point can influence the distribution and abundance of invasive worms and these effects may be best understood in the context of campsite use behaviors and bait use practices. We recommend that resource managers promote “leave no trace” principles that emphasize the environmental risks posed by invasive species; for example, by incorporating this message in videos visitors are required to watch before entering the BWCAW (Guo et al., 2017). As part of their public education efforts, managers might also consider bait container warning labels that include instructions on the proper earthworm disposal after use. These actions may slow the spread of earthworms in the BWCAW and other wilderness areas, although removing these ecosystem engineers from invaded regions is problematic at best, if not impossible. Once introduced, earthworms become part of the ecosystem (Bohlen et al., 2004b) and efforts directed at learning ways to mitigate their impacts will likely become as important as trying to control their spread (Tammeorg et al., 2014).

Supplemental Information

Supplemental Information 1 Earthworm data collected from BWCA islands, 2016.

Physical characteristics include island latitude & longitude, island area (m2), distance from entry point (m), island distance from lake shore, and whether island was burned and had campsites, island distance from lake shore (m). Also included are measured variables: earthworm density (#s/m2) and biomass (g/m2) and species richness; and data from sample sites: ground cover (%) understory vegetation cover (%) and soil organic matter (%).

Click here for additional data file.

Supplemental Information 2 Supplemental files showing analysis of residuals for SEM.

Figure 1: Residuals from the SEM compared across the lakes. Figure 2: Variogram of residuals from the SEM. The plot shows no evidence for spatially auto-correlated residuals, indicating samples from same lakes can be treated as independent.

Click here for additional data file.

We wish to thank Wil Raasch for his contributions to the work, and David Lonzarich for his assistance with the SEM analysis.

Additional Information and Declarations

Competing Interests

Author Contributions

Field Study Permissions

Data Availability

Todd Wellnitz has previously been a reviewer for PeerJ. The other authors have no competing interests.

Todd Wellnitz conceived and designed the experiments, performed the experiments, analyzed the data, prepared figures and/or tables, authored or reviewed drafts of the paper, and approved the final draft.

Jenna L. Barlow conceived and designed the experiments, performed the experiments, analyzed the data, prepared figures and/or tables, authored or reviewed drafts of the paper, and approved the final draft.

Cory M. Dick conceived and designed the experiments, performed the experiments, analyzed the data, prepared figures and/or tables, authored or reviewed drafts of the paper, and approved the final draft.

Terrance R. Shaurette conceived and designed the experiments, performed the experiments, analyzed the data, prepared figures and/or tables, authored or reviewed drafts of the paper, and approved the final draft.

Brian M. Johnson performed the experiments, prepared figures and/or tables, authored or reviewed drafts of the paper, and approved the final draft.

Troy Wesley performed the experiments, prepared figures and/or tables, and approved the final draft.

Evan Weiher analyzed the data, authored or reviewed drafts of the paper, and approved the final draft.

The following information was supplied relating to field study approvals (i.e., approving body and any reference numbers):

United State Department of Agriculture, United States Forest Service, Boundary Waters Canoe Area Wilderness provided permit #6-2861367 (issued 8 July 2016).

The following information was supplied regarding data availability:

Data is available at the The Knowledge Network for Biocomplexity (KNB): Todd Wellnitz. 2016. 2016 Earthworm data from the Boundary Waters Canoe Area Wilderness, Minnesota, USA. Knowledge Network for Biocomplexity. urn:uuid:ded738fa-e662-488c-8d34-f4e8660c8912.

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
