# Peer review of "Campsites, forest fires, and entry point distance affect earthworm abundance in the Boundary Waters Canoe Area Wilderness"

_PeerJ, doi:10.7717/peerj.8656_

## Round 0.1 · original submission · Major Revisions

I haven't sent this paper out for review yet, because I think that you need to assess the resilience of your conclusions to the assumptions you've made in the analysis.

You note that you decided not to use nested analysis because the lakes are not unequivocally defined. I don't find that a sufficiently compelling reason, especially considering that you do give the lakes numbers. Please extend your analysis to demonstrate whether or not it really matters. You can do that either by (i) fitting a hierarchical model and repeating the inference or (ii) careful analysis of the residuals - e.g., is the intra-class correlation (or equivalent) arguably negligible for the plausible candidate lake classifications?

Finally, my apologies for the delay in handling; it turns out that PeerJ isn't really set up for the AE to decide to return the paper without soliciting review comments.

---

## Round 0.2 · Minor Revisions

The review comments are pithy and reasonable. Please consider all of them.

Reviewer 1 ·

Basic reporting

I commend the authors, as this paper is well written with sufficient background, results, and discussion. However, there are some minor edits that I recommend:

- There is inconsistency in the use of "Boundary Waters Canoe Area", "Boundary Waters", "Boundary Waters wilderness", "BWCA", etc. to refer to the study site. I believe that the official name is Boundary Waters Canoe Area Wilderness (BWCAW). The authors should use the full name at first mention, introducing the acronym in parentheses, and thereafter use the acronym.

- Line 68-70, This sentence is awkward, suggest rephrasing.

- Line 78, Here you use epigeic but later you use epigenic (Line 284). Keep consistent.

- Line 119, Need to specify if this citation is 2017a or 2017b

- Line 192, Delete A and start paragraph with Structural Equation Modelling....

- Line 211, Correct Poison to Poisson

- Line 267-269, See also Shartell et al. 2015 for earthworm relationships to anthropogenic activity.

- Line 286, Missing a word? Need to fix this sentence

- Line 290, Insert the word "on" after effects

- Line 339-342, This is the opinion of the authors and I don't believe this citation should be used. The cited paper is not on earthworm removal (see comment above, this cited paper is related to anthropogenic use and earthworms). You could search the literature for research related to removing earthworms. For example, Madritch and Lindroth. 2009. Removal of invasive shrubs reduces exotic earthworm populations. Biological Invasions.

Experimental design

The question and methods are well defined but how it is relevant to management is not fully addressed. Why is this information meaningful and how can the results be used to improve conservation and management of this wilderness area?

Validity of the findings

I believe the findings are valid and interesting.

Additional comments

Nice work. I think this paper is ready for publication with the minor edits above. I had a few thoughts that do not need to be included in this paper, but could spark future work.

What is the probability of multiple introductions on earthworm metrics. What anthropogenic features are more likely to lead to multiple introductions, and thus effect richness, etc.

I was also curious about the understory plant spp diversity, and whether the plants were mainly native or if exotics/invasives were present.

Do these islands have a past history of fire (prior to Pagami Creek Fire)? How might that have an effect on earthworms? Is the reduction of earthworms after fire temporary? How long do fire effects last?

Reviewer 2 ·

Basic reporting

The paper meets all of these standards.

Experimental design

The paper meets all of these standards

Validity of the findings

The paper also meets these standards

Additional comments

This is a nicely done study with interesting results. There aren’t any major flaws in the design, data collection of analyses or interpretation. The number of samples per island, and size of each sample on the ground, the way samples were handled in the lab all seem well founded. It is good that they measured both density and biomass of earthworms. I am not surprised that islands within lakes turned out to be independent for purposes of the study—that is in agreement with what I have found in the BWCAW. Finally, explaining almost half the variance in a SEM model is quite good for this type of study. The positive campsite effect on worm biomass seems very reasonable and well supported by the data, with a p-value of 0.001 in GLM analysis. I think the explanation given in the discussion for why earthworm biomass increases further into the wilderness, is reasonable. It does relate to human behavior and use of live bait. I leave it to the authors to decide if they want to make any revisions to the discussion in response to points 1 and 2 below, and following that there are some minor editorial items numbered 3, 4, and 5:

1. The study captured some of the effects of the fire, 4.5 growing seasons after it occurred, and obviously those effects would have been more prominent 1-2 years after the fire and will gradually disappear with time since fire. However, the snapshot after 4.5 growing seasons still contains useful information given how little we know about fire and earthworm interactions.

2. Regarding the discussion about the positive effect of worms on understory plants (lines 292-299). The the explanation might be that the study is detecting places where environment was good for both plants and worms—in other words pockets of deeper soil on this rocky landscape, and that SEM can’t really show those two-way relationships (in fact, I don’t know of any statistical methods that can do that).

Minor comments:
3. Line 130, 40% is not ‘mostly’
4. Line 192, reword, either say 'A Structural Equation Model was used...' or 'Structural Equation Modelling was used...'
5. Line 290—missing word—‘on these variables’?

---

## Round 0.3 · accepted · Accept

Thanks for your careful handling of all the comments.